# Physician-Specific Symptoms of Burnout Compared to a Non-Physicians Group

**DOI:** 10.3390/ijerph20032693

**Published:** 2023-02-02

**Authors:** Hermanas Usas, Sonja Weilenmann, Mary Princip, Walther J. Fuchs, Marc van Nuffel, Roland von Känel, Tobias R. Spiller

**Affiliations:** 1Department of Consultation-Liaison Psychiatry and Psychosomatic Medicine, University Hospital Zurich (USZ), Haldenbachstrasse 16/18, CH-8091 Zurich, Switzerland; 2Faculty of Medicine, University of Zurich (UZH), CH-8006 Zurich, Switzerland; 3Burnout Protector GmbH, CH-8700 Küsnacht, Switzerland; 4Digiboo® GmbH, CH-8700 Küsnacht, Switzerland; 5DU DA—Data & Commtech by Farner, CH-8004 Zürich, Switzerland

**Keywords:** burnout, physicians, fatigue, work stress, online survey

## Abstract

Physician burnout is a systemic problem in health care due to its high prevalence and its negative impact on professional functioning and individual well-being. While unique aspects of the physician role contributing to the development burnout have been investigated recently, it is currently unclear whether burnout manifests differently in physicians compared to the non-physician working population. We conducted an individual symptom analysis of burnout symptoms comparing a large sample of physicians with a non-physician group. In this cross-sectional online study, burnout was assessed with the Maslach Burnout Inventory—General Survey. We matched physicians with non-physicians regarding their age, gender, educational level, occupational status, and total burnout level using a “nearest neighbour matching” procedure. We then conducted a series of between-groups comparisons. Data of 3846 (51.0% women) participants including 641 physicians and 3205 non-physicians were analysed. The most pronounced difference was that physicians were more satisfied with their work performance (medium effect size (r = 0.343). Our findings indicate minor yet significant differences in burnout phenomenology between physicians and non-physicians. This demonstrates unique aspects of physician burnout and implies that such differences should be considered in occupational research among physicians, particularly when developing burnout prevention programs for physicians.

## 1. Introduction

Burnout was first conceptualized in the 1970s to describe work stress-induced changes in employees of health care facilities [1]. Since then, research has demonstrated that burnout affects all working populations [2]. However, it is likely not a coincidence that burnout was first described in health care professionals, given a prevalence of 40% or more of burnout among physicians [3,4,5]. Importantly, burnout has been shown to be related to multiple negative health outcomes, including mental disorders [6] and physical diseases [7]. In addition, compared to their counterparts without burnout, physicians with burnout report that they make more medical errors [8] and provide suboptimal patient care [9].

Although not considered a mental disorder, burnout is listed in the 10th international classification of diseases (ICD-10) as an “occupational phenomenon” [10]. In the coming 11th edition of the ICD, the burnout syndrome is defined as (emotional) exhaustion, mental distancing or feeling of negativism and reduced professional efficacy [11]. This definition closely follows the three-dimensional operationalization of burnout first established by Maslach and Jackson [12] and is among the most prominent operationalizations of burnout, although not the only one [13]. Notably, these three dimensions are commonly termed emotional exhaustion (EE), depersonalization (DP), and professional efficacy (PE). The prevalence of each dimension can differ substantially in the same population. For example, in one large sample of U.S. physicians, 46.9% reported a high score for EE, whereas high scores for PE were only found in 34.6% [14]. Furthermore, the three dimensions are also differently related to mental disorders and physical diseases [6]. For example, emotional exhaustion was found to be associated more strongly with symptoms of depression [15], somatisation [16], and hypertension [17] than other dimensions of burnout. Therefore, differences in the phenomenology of burnout, i.e., the combination of symptoms of burnout an individual displays, can lead to differences in risk regarding the above-mentioned negative health outcomes. This is important given the high overlap between burnout with anxiety disorders and depression, which both negatively impact personal well-being as well as job performance [18,19]. Thus, a better understanding of the phenomenology of burnout in physicians might help to better understand its association with depression and anxiety, ultimately paving the path for better prevention measures.

In burnout research, physicians are likely the most well-studied population and one with a substantially higher prevalence of burnout than the general working population [3,20]. The higher prevalence of burnout among physicians has been attributed in part to the higher presence of known risk factors for burnout among physicians, e.g., excessively long working hours [21]. In addition, some have suggested that physician burnout is also the result of factors unique to the professional role of physicians, e.g., the role physicians have in health care organizations [21,22]. While differences in the prevalence between physicians and the general population have been systematically studied, differences in phenomenology have largely been neglected to date. However, there is indirect evidence that physician burnout differs phenomenologically from burnout in other populations. For example, O`Connor et al. reported in a meta-analysis of burnout among mental health professionals (including physicians, among others) that EE was more prevalent than DP, which was again more prevalent than low PE in mental health professionals [23]. This is in contrast to a meta-analysis of burnout among secondary school teachers, which reported a higher prevalence of DP than EE [24]. Similarly, a single study of South African police officers reported higher mean scores for PE than for EE and DP [25]. Notably, only one study directly compared differences in burnout symptoms between physicians and non-physicians. In this study, Shanafelt et al. reported markedly higher rates of EE, DP and overall prevalence of burnout in physicians compared to the general population [14]. However, in their study, the physician and the non-physician groups differed in many regards. Among others, a larger share of the physicians were males and reported being married; they were also, on average, older and worked more hours per week than the sample from the general U.S. population [14]. Thus, it is unclear whether the found differences represent true differences in the symptomatic manifestation of burnout or are the result of differences in the groups’ characteristics.

Therefore, we aimed to examine phenomenological differences between physicians and a non-physician comparison group on a single symptom basis. To adjust for differences in individual symptoms due to differences in demographics or burnout severity, we matched physicians with non-physicians using propensity score matching [26]. By doing so, we aim to elucidate whether physician burnout is phenomenologically distinct from burnout in a non-physician comparison group of the working population. Such potential differences could indicate that physician burnout has unique aspects that need to be considered when working with this population.

## 2. Material and Methods

### 2.1. Procedure and Participants

This cross-sectional study was carried out as an online study. Participants self-reported their demographics, and their symptoms of burnout were assessed with the Maslach Burnout Inventory—General Survey (MBI-GS). This was part of an online self-assessment of burnout risk, which was assessable for free online and hosted on a web-page belonging to a private company called Burnout Protector© [27]. This online assessment was advertised with a campaign launched via Medinside, Winsider AG, Winterthur, Switzerland, an open-access publication covering news in the Swiss health care sector. Data of participants who used this online assessment between November 2016 and September 2019 were considered for this study. All participants younger than 18 years and older than 70 years, the latter being the age of the latest official retirement in Switzerland, were excluded from the analysis. Due to the nature of the online study, participants could not be prevented from participating more than once. To minimize bias (e.g., due to participants exploring how different answers change the overall burnout score) and to include individuals only once (i.e., assigning every participant the same weight in the sample), only the data of the first assessment were considered in participants who used the online application more than once. Hence, no sample size calculations were undertaken but all available data of participants satisfying the above-mentioned inclusion criteria were analysed in this study. 

This study did not fall within the scope of the Swiss Human Research Act. Therefore, no authorization from an Ethics committee was required. Still, all participants were asked for their informed consent to use their anonymized data for scientific purposes. Moreover, this study was conducted in accordance with the Swiss Human Research Ordinance, the Swiss Federal Act on Research Involving Human Beings [28] and the guidelines relevant for this study issued by the Swiss Federal Data Protection Commissioner [29].

### 2.2. Measures

Demographics. Data on demographic variables included age (in years), gender (male or female), highest level of education (university degree, vocational school certificate, high school degree or junior middle school degree) and being a physician or not.

Maslach Burnout Inventory—General Survey. Burnout symptoms were assessed with the Maslach Burnout Inventory—General Survey (MBI-GS) [30]. Each of the 16 items is rated on a 7-point Likert scale ranging from 0 (“never”) to 6 (“daily”) covering the occurrence of each burnout symptom during the last twelve months. The MBI-GS consists of three dimensions, namely EE, DP, and PE. Overall burnout severity is calculated as a weighted sum score of the three dimensions. This is carried out by first summing the items of each dimension and dividing them by the number of items per dimension, weighting the sum scores by multiplying them with 0.4 (for EE) or 0.3 (for DP and PE) and then adding the three scores to the overall sum score [31]. Thus, this overall sum score ranges from 0 to 6. The internal consistency of the MBI-GS was reported to be satisfactory with a lower limit of Cronbach’s alpha of 0.70 for PE and an upper limit of 0.90 for EE [2]. In this study, Cronbach’s alpha for EE, DP, PE and overall was 0.91, 0.84, 0.72, and 0.89, respectively.

### 2.3. Data Analysis

All data analyses were carried out in the *R* statistical environment [32] using additional established “packages” that provide additional functionality and analyses.

Matching. Following recommendations outlined for the use of matching procedures in observational studies [33], the matching of physicians with non-physicians was conducted using the “nearest neighbour matching” procedure [26]. Only complete data can be handled by this procedure. Therefore, participants with missing data in one of the matching variables were excluded from further analysis. Participants were then matched regarding their age, gender, level of education, status as a physician or not and overall burnout score, with a one-to-five ratio meaning that each physician was matched to five non-physicians. We used this matching procedure as implemented in the *R* package *MatchIt* [34], as it is has been extensively validated and is among the most commonly used packages that implement matching procedures [35].

Descriptive statistics and group differences. The description of categorical variables was provided with frequencies (%) and the two groups (physicians and non-physicians) were compared using chi-squared tests. Continuous variables were described using the mean and standard deviation. Independent group comparisons were conducted using two-tailed Mann–Whitney U tests. The significance level was set to alpha = 0.05. *p* values for the individual item comparisons were adjusted for multiple comparisons using the Benjamini–Hochberg procedure [36]. Effect sizes were assessed using the Rank-biserial correlation coefficient [37]. 

## 3. Results

### 3.1. Participants

The characteristics of the 3846 participants are presented in Table 1. The 641 physicians had a mean age of 40.9 years (SD—11.9) and the 3205 non-physician participants had a mean age of 41.0 years (SD—11.7). The gender distribution was similar in both groups (50.7% each in the non-physician, and 51.0% females in the physician group). Burnout total severity score was comparable in both groups, with a value of 2.15 (SD—1.13) in the group of physicians and 2.16 (SD—1.18) in the non-physician comparison group.

### 3.2. Phenomenology of Burnout

The comparison of both groups on burnout symptoms is shown in Table 2. After the adjustment of the *p* value for multiple comparisons, there were significant differences in 9 out of 16 MBI-GS items. Regarding the magnitude of these differences, the largest difference was found for the item *“Being good in the job.”* (PE3), with physicians reporting to think this way more often than non-physicians (effect size: r = 0.343). The effect size for all other differences was below r = 0.1, indicating only very small effects. A graphical representation of the results, outlining the mean and the standard error for each of the MBI-GS’s items is presented in Figure 1.

EE1-5—emotional exhaustion items 1 to 5; PE1-6—personal efficacy items 1 to 6; DP1-5—cynicism items 1 to 5; Note that higher values for burnout items indicate more frequent occurrence of a particular symptom (Table 3). Data are shown as means with SD.

## 4. Discussion

This study investigated differences in symptoms of burnout between physicians and a non-physician comparison group using a case matching design, controlling for age, gender, highest level of education and overall burnout severity. We found several significant group differences in the severity of individual burnout symptoms. These differences were small with one important exception: physicians generally reported that they more frequently think that they are good at their job than non-physicians did.

The higher ratings of physicians of their own job performance are interesting and relevant. Interestingly, the other items of the PA subscale primarily assess the outcome of one’s work or one’s attitude towards these outcomes (e.g., PE1 “Effectively solving work problems.” or PE4 “Feeling exhilarated by work accomplishments.”). Thus, physicians affected by burnout report a loss of efficacy and value their achievements less; however, they still think that they are doing a good job. This seemingly paradoxical finding might be explained by the fact that physicians consider these losses a consequence of the working conditions in the health system, rather than a result of personal failure. This concurs with the recent change in focus from personal to systemic causes for burnout in physicians [38,39]. Moreover, a mismatch between effort and result, as well as inadequate administrative load, might be an important factor in developing burnout. Cynicism can be seen as a strategy to cope with dwindling effort-associated rewards and perceived powerlessness. Moreover, cynicism has been associated with other dysfunctional coping mechanisms including disengagement with work, the intention to leave the profession, and depression [40]. In addition to the institutional factors presented by Shanafelt and Noseworthy et al. [41], presenteeism culture [42,43] might hinder the necessary steps needed to combat burnout. This can, in turn, make leaving the profession altogether the only viable route to escape the external influences that contribute to burnout. Further research should concentrate on identifying individuals at risk and specific institutional risk factors in burnout development to guide implementation of effective policies oriented at maintaining physicians’ wellbeing.

Our finding that physicians and non-physicians reported predominantly similar burnout phenomenology is in contrast with those of several previous studies [3,14,20]. This difference might be explained by differences in baseline characteristics between physicians and non-physicians that were not accounted for in previous studies [3,14,20,39] but in our present one. Given that the two investigated populations showed only minor disparity in a single questionnaire item after the adjustment of baseline differences, future studies comparing burnout between professions should adjust for baseline differences.

This study is subject to several limitations. Firstly, we analysed data that were collected from a non-representative sample. Therefore, our results are explorative. Future research should aim to replicate our results in an independent and preferably representative dataset. Secondly, no detailed information about the jobs performed by the comparison group was collected, which limits the external validity of our findings. Nevertheless, participants’ educational level was not only assessed but it was also included in the matching score. Thus, all the included participants had a formal education comparable to that of the physicians (i.e., university degree). Thirdly, the data collection process using a freely accessible web page might have introduced biases, for example, by increasing the likelihood of someone concerned about burnout participating in the study. This limits the generalizability of our findings to the wider physician and working populations. Future research should compare the differences in burnout symptoms when these data are gathered online versus offline. Fourthly, although the internal consistency of the MBI-GS was very good in this study, its broader reliability (e.g., test–re-test reliability) was not assessed but should be undertaken by future studies to ensure the validity of the findings. Finally, the study was conducted in multiple languages, including English. Therefore, although the study was promoted in Switzerland only, participants from all over the globe could have participated in this study. Hence, the generalizability to other countries is currently unclear.

## 5. Conclusions

This study aimed to assess differences in individual burnout symptoms between a physician and a non-physician professional group using data collected online. We matched physicians with non-physicians based on their demographics and burnout severity to minimize bias using propensity score matching. While physicians’ perceived loss of efficacy and appreciation for their work was comparable to that of non-physicians, they reported that they think they do a good job more often. This indicates that physicians, in contrast with the other working population, tend to preserve a positive self-image of their work even when other symptoms of burnout are present. Future work should aim to elaborate in which specific aspects of the role of the physician these differences are bound to be present.

## Figures and Tables

**Figure 1 ijerph-20-02693-f001:**
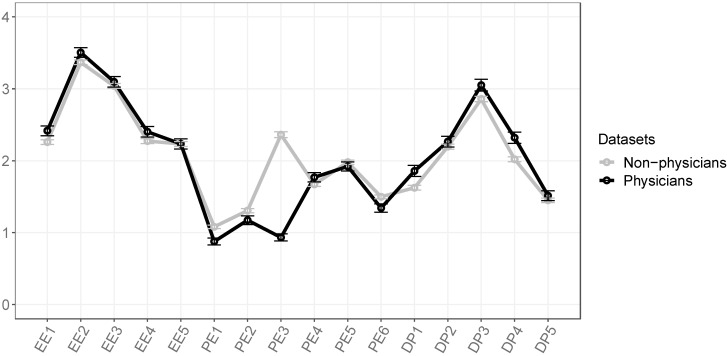
Mean burnout symptoms of physicians and non-physicians.

**Table 1 ijerph-20-02693-t001:** Demographics.

Variable	Non-Physicians	Physician	Overall
	(n = 3205)	(n = 641)	(n = 3846)
Age			
Mean (SD)	41.0 (11.7)	40.9 (11.9)	41.0 (11.7)
Gender			
Male	1580 (49.3%)	314 (49.0%)	1894 (49.2%)
Female	1625 (50.7%)	327 (51.0%)	1952 (50.8%)
Burnout severity			
Mean (SD)	2.16 (1.19)	2.15 (1.13)	2.16 (1.18)

**Table 2 ijerph-20-02693-t002:** Differences in burnout symptoms of physicians and non-physicians.

Abbreviation	Non-Physician	Physician	Overall	*p*	Effect Size (r)
	(n = 3205)	(n = 641)	(n = 3846)		
EE1	2.26 (1.87)	2.42 (1.71)	2.29 (1.84)	<0.001	0.073
EE2	3.37 (1.83)	3.50 (1.71)	3.39 (1.81)	0.037	0.039
EE3	3.04 (1.95)	3.10 (1.84)	3.05 (1.93)	0.835	0.004
EE4	1.67 (1.63)	1.77 (1.64)	1.69 (1.63)	0.004	0.054
EE5	2.23 (1.89)	2.23 (1.79)	2.23 (1.87)	0.341	0.018
PE1	1.08 (1.36)	0.877 (1.23)	1.05 (1.34)	<0.001	0.069
PE2	1.31 (1.59)	1.17 (1.54)	1.28 (1.59)	0.669	0.008
PE3	2.36 (2.30)	0.933 (1.26)	2.12 (2.23)	<0.001	0.261
PE4	1.67 (1.63)	1.77 (1.64)	1.69 (1.63)	0.004	0.054
PE5	1.98 (1.70)	1.92 (1.63)	1.97 (1.69)	0.044	0.038
PE6	1.50 (1.51)	1.34 (1.46)	1.47 (1.51)	0.437	0.015
DP1	1.62 (1.90)	1.86 (1.94)	1.66 (1.91)	<0.001	0.096
DP2	2.20 (1.94)	2.27 (1.92)	2.21 (1.94)	0.018	0.045
DP3	2.86 (2.10)	3.05 (2.06)	2.89 (2.09)	0.174	0.026
DP4	2.02 (1.94)	2.32 (1.97)	2.07 (1.95)	<0.001	0.093
DP5	1.45 (1.79)	1.51 (1.75)	1.46 (1.78)	0.002	0.056

Notes: All numerical values except *p* are represented with mean (SD) value; The *p*-values shown were adjusted for multiple comparisons; EE1-5—emotional exhaustion items 1 to 5; PA1-6—personal efficacy items 1 to 6; DP1-5—depersonalization items 1 to 5. Note that higher values for burnout items indicate more frequent occurrence of a particular symptom.

**Table 3 ijerph-20-02693-t003:** Legend.

Abbreviation	Item Number	Question
EE1	1	Feeling emotionally drained.
EE2	2	Feeling used up.
EE3	3	Feeling tired before another workday.
EE4	4	Feeling strained by work.
EE5	6	Feeling burned out from work.
PE1	5	Effectively solving work problems.
PE2	7	Making a contribution to the organization.
PE3	10	Being good in the job.
PE4	11	Feeling exhilarated by work accomplishments.
PE5	12	Feeling of accomplishment.
PE6	16	Feeling efficacious.
DP1	8	Losing interest in one’s work.
DP2	9	Feeling less enthusiastic.
DP3	13	Feeling distanced from work.
DP4	14	Feeling cynical about work.
DP5	15	Doubting the significance of one’s work.

Notes: EE1-5—emotional exhaustion items 1 to 5; PE1-6—personal efficacy items 1 to 6; DP1-5—depersonalization items 1 to 5.

## Data Availability

The anonymized datasets analysed during the current study are available from the corresponding author on reasonable request.

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
