# Peer review of "Physician-Specific Symptoms of Burnout Compared to a Non-Physicians Group"

_ijerph, 2023, doi:10.3390/ijerph20032693_

Round 1
Reviewer 1 Report
This is an interesting study and relevant for the field study, especially for not a physician or health care professional.
The followings suggestions and corrections are recommended:
1. There are some mixed-up citation styles according to the journal's citation style. Lines 66-67
2. Authors must mention clearly which ones were studied in their research and why? Line 76
3. Procedure and Participants, this part of the procedure must be clarified. The corresponding results of this part are not well explained in the results section. It Is confusing if this online assessment actually uses the Maslach approach.
4. Lines 96-97 Authors must clarify why this decision was made and how its affects results.
5. Lines 119-122, this part of the study must be clarified since earlier in the manuscript refer to the Protector Online package. Please, explain how this online package uses the Maslack approach.
6. Lines 124-125 this part of the study needs theoretical support and why this package was used among others. An extension of the Literature review of this method will help support its use.
7. What can you conclude about the main objective of the study, other relevant conclusions about the methods used, the online applications used, its reliability are very important, Lines 210-214

Author Response
This is an interesting study and relevant for the field study, especially for not a physician or health care professional.
We thank the reviewer for their helpful comments and suggested recommendations. Please find a response to detailed response below.
The followings suggestions and corrections are recommended:
1. There are some mixed-up citation styles according to the journal's citation style. Lines 66-67
Reply: We agree with the reviewer that there were deviations from the journal’s citation style. We did correct them using Zotero and the style template provided by the publisher. This has led to the following changes.
Lines 66-67: For example, O`Connor et al. reported in a meta-analysis of burnout among mental health professionals (including among others physicians) that EE was more prevalent than DP, which was again more prevalent than low PE in mental health professionals [18].
Lines 73-75: Shanafelt et al. reported markedly higher rates of EE, DP and overall prevalence of burnout in physicians compared to the general population [14].
- Authors must mention clearly which ones were studied in their research and why? Line 76
Reply: We did provide more details about the differences of the two sample in the study by Shanafelt et al.
Lines 75-77: Shanafelt et al. reported markedly higher rates of EE, DP and overall prevalence of burnout in physicians compared to the general population [14]. However, in their study the physician and the non-physician groups differed in many regards. Among others, a larger share of the physicians were males, reported being married, they were also on average older and worked more hours per week than the sample from the general U.S. population [14].
- Procedure and Participants, this part of the procedure must be clarified. The corresponding results of this part are not well explained in the results section. It Is confusing if this online assessment actually uses the Maslach approach.
Reply: Thank you very much for raising this issue. The description of the procedure was not as clear as it could have been, and we think this has led to some confusion. The study was conducted online using the Maslach Burnout Inventory – General Survey. The webpage on which the questionnaires were hosted belongs to a company called “Burnout Protector”. We have clarified the paragraph to unambiguously state the role of “Burnout Protector” and adjusted the manuscript accordingly.
Lines 89-92: This cross-sectional study was carried out as an online study. Participants self-reported their demographics, and their symptoms of burnout were assessed with the Maslach Burnout Inventory – General Survey (MBI-GS). This was part of an online self-assessment of burnout risk, which was assessable for free online and hosted on a web-page belonging to a private company called Burnout Protector©; [21]. This online assessment was advertised with a campaign launched via Medinside, Winsider AG, Winterthur, Switzerland, an open access publication covering news in the Swiss health care sector. Data of participants who used this online assessment between November 2016 and September 2019 were considered for this study.
- Lines 96-97 Authors must clarify why this decision was made and how it affects results.
Reply: We extended this sentence to provide a rationale for this choice.
Lines 93-94: Due to the nature of the online study, participants could not be prevented from participating more than once. To minimize bias (e.g., due to participants exploring how different answers change the overall burnout score) and to include individuals only once (i.e., assigning every participant the same weight in the sample), only the data of the first assessment was considered in participants who used the online application more than once.
- Lines 119-122, this part of the study must be clarified since earlier in the manuscript refer to the Protector Online package. Please, explain how this online package uses the Maslach approach.
Reply: We are sorry that the provided description was confusing. “Burnout Protector” is the name of the company that hosted the self-assessment with the MBI-GS. “MatchIt” is a “package” in the R environment – a statistical language similar to SPSS or Python. “MatchIt” implements the used propensity score matching methods. To clarify this, we have made the following changes to the manuscript:
Introduction: Therefore, we aimed to examine phenomenological differences between physicians and a non-physician comparison group on a single symptom basis. To adjust for differences in individual symptoms due to differences in demographics or burnout severity, we matched physicians with non-physicians using propensity score matching. By doing so, we aim to elucidate whether physician burnout is phenomenologically distinct from burnout in a non-physician comparison group of the working population. Such potential differences could indicate that physician burnout has unique aspects that need to be considered when working with this population.
Methods – Procedure: This cross-sectional study was carried out as an online study. Participants self-reported their demographics, and their symptoms of burnout were assessed with the Maslach Burnout Inventory – General Survey (MBI-GS). This was part of an online self-assessment of burnout risk, which was assessable for free online and hosted on a web-page belonging to a private company called Burnout Protector©; [21]. This online assessment was advertised with a campaign launched via Medinside, Winsider AG, Winterthur, Switzerland, an open access publication covering news in the Swiss health care sector.
Methods – Data Analysis:
Data Analysis
All data analyses were carried out in the R statistical environment [29] using additional established “packages” that provide additional functionality and analyses.
Matching. Following recommendations outlined for the use of matching procedures in observational studies, matching of physicians with non-physicians was conducted using the “Nearest Neighbor Matching” procedure. Only complete data can be handled by this procedure. Therefore, participants with missing data in one of the matching variables were excluded from further analysis. Participants were then matched regarding their age, gender, level of education, status as a physician or not and overall burnout score, with a one-to-five ratio meaning that each physician was matched to five non-physicians. We used this matching procedure as implemented in the R package MatchIt [26], as it is has been extensively validated and is among the most commonly used packages that implemented matching procedures.
- Lines 124-125 this part of the study needs theoretical support and why this package was used among others. An extension of the Literature review of this method will help support its use.
While SPSS and similar programs only offer one option for an analysis, the same analytic procedure can be implemented in different R packages. We have chosen to use “MatchIt” because it is the most commonly used package for this kind of analysis (cited more than 3000 times) and has been validated extensively. We have adjusted the manuscript accordingly.
Lines 125-126: Matching. Following recommendations outlined for the use of matching procedures in observational studies, matching of physicians with non-physicians was conducted using the “Nearest Neighbor Matching” procedure. Only complete data can be handled by this procedure. Therefore, participants with missing data in one of the matching variables were excluded from further analysis. Participants were then matched regarding their age, gender, level of education, status as a physician or not and overall burnout score, with a one-to-five ratio meaning that each physician was matched to five non-physicians. We used this matching procedure as implemented in the R package MatchIt [26], as it is has been extensively validated and is among the most commonly used packages that implemented matching procedures.
- What can you conclude about the main objective of the study, other relevant conclusions about the methods used, the online applications used, its reliability are very important, Lines 210-214
Reply: We agree with the reviewer that the main conclusion should be based upon reliable evidence. Given the matching procedure and the large size of the sample, we are confident that we can make inferences about the main objective of the study – a symptom level comparison of the symptoms of burnout between physicians and non-physicians. However, we agree that we could have been clearer about this objective and have expanded limitations and adjusted the conclusion accordingly.
Limitations: This study is subject to several limitations. Firstly, we analysed data that were collected from a non-representative sample. Therefore, our results are explorative. Future research should aim to replicate our results in independent and preferably representative dataset. Second, no detailed information about the jobs performed by the comparison group was collected which limits the external validity of our findings. Nevertheless, participants educational level was not only assessed but it was also included into the matching score. Thus, all the included participants had a formal education comparable to the physicians (i.e., high academic degree). Third, the data collection using a freely accessible web page might have introduced biases, for example by increasing the likelihood of someone concerned about burnout to participate in the study. This limits the generalizability of our findings to the wider physician and working population. Future research should compare the differences in burnout symptoms when these data are gathered online versus offline. Fourth, although the internal consistency of the MBI-GS was very good in this study, its broader reliability (e.g., test-re-test reliability) was not assessed but should be undertaken by future studies to ensure the validity of the findings. Finally, the study was conducted in multiple languages, including English. Therefore, although the study was promoted in Switzerland only, participants from all over the globe could have participated in this study. Hence, the generalizability to other countries is currently unclear.
Conclusion: This study aimed to assess differences in individual burnout symptoms between a physician and a non-physician professional group using data collected online. We matched physicians with non-physicians based on their demographics and burnout severity to minimize bias using propensity score matching. While physicians’ perceived loss of efficacy and appreciation for their work was comparable to non-physicians, they report more often that they think they do a good job. This indicates that physicians, in contrast to the other working population, tend to preserve a positive self-image of their work even when other symptoms of burnout are present. Future work should aim to elaborate to which specific aspect of the role of the physician these differences are bound.
Reviewer 2 Report
Thank you for submitting the manuscript. I have read your paper with great attention and interest. The subject is certainly of interest and of great relevance. The idea is good and there are excellent ideas, both in terms of results and in terms of prospects for future research.However, I have to ask for revisions. In fact, in the introduction, burnout is treated as an isolated phenomenon, when instead it is often accompanied by psychosomatic manifestations or other psychological conditions. In this regard, I suggest the following references that I would like you to read and use:doi: 10.13075/ijomeh.1896.01323. DOI: 10.3390/healthcare10081370 doi: 10.5546/aap.2021.eng.317.
I also ask you to specify the type of work performed by the sample which is compared to health workers. This is very important information, as burnout presents different conditions depending on the type of job. I hope my comments are helpful to you
Kind Regards
Author Response
Reviewer 2
Thank you for submitting the manuscript. I have read your paper with great attention and interest. The subject is certainly of interest and of great relevance. The idea is good and there are excellent ideas, both in terms of results and in terms of prospects for future research. However, I have to ask for revisions.
In fact, in the introduction, burnout is treated as an isolated phenomenon, when instead it is often accompanied by psychosomatic manifestations or other psychological conditions. In this regard, I suggest the following references that I would like you to read and use: doi:10.13075/ijomeh.1896.01323. DOI: 10.3390/healthcare10081370 doi: 10.5546/aap.2021.eng.317.
Reply: The reviewer is absolutely correct that burnout is strongly associated with depression and anxiety (and other psychological conditions). This was indeed not mentioned, and we have adjusted the introduction accordingly using the suggested references.
Lines 45-55: The prevalence of each dimension can differ substantially in the same population. For example, in one large sample of U.S. physicians, 46.9% reported a high score for EE whereas high scores for PE were only found in 34.6% [14]. Furthermore, the three dimensions are also differently related to mental disorders and physical diseases [6]. For example, emotional exhaustion was found to be associated more strongly with symptoms of depression, somatisation ref, and hypertension REF than other dimensions of burnout. Therefore, differences in the phenomenology of burnout, i.e., the combination of symptoms of burnout an individual displays, can lead to differences in risk regarding the above-mentioned negative health outcomes. This important given the high overlap between burnout with anxiety disorders and depression, which both negatively impact personal well-being as well as job performance. Thus, a better understanding of the phenomenology of burnout in physicians might help to better understand its association with depression and anxiety, ultimately paving the path for better prevention measures.
I also ask you to specify the type of work performed by the sample which is compared to health workers. This is very important information, as burnout presents different conditions depending on the type of job.
Reply: We agree with the reviewer, that the kind of work performed by the comparison group is relevant information. Unfortunately, we only have information about the educational level of the participants but no detailed information about their current job situation. The individuals were matched for their educational level, which should minimize bias. Nevertheless, this is an important limitation and we have added it to the corresponding section.
Limitations: This study is subject to several limitations. Firstly, we analysed data that were collected from a non-representative sample. Therefore, our results are explorative. Future research should aim to replicate our results in independent and preferably representative dataset. Second, no detailed information about the jobs performed by the comparison group was collected which limits the external validity of our findings. Nevertheless, participants educational level was not only assessed but it was also included into the matching score. Thus, all the included participants had a formal education comparable to the physicians (i.e., high academic degree).
I hope my comments are helpful to you
Reply: We thank the reviewer for their helpful comments. We believe they made the paper significantly stronger.